# Clinical Significance of Elevated Xanthine Dehydrogenase Levels and Hyperuricemia in Patients with Sepsis

**DOI:** 10.3390/ijms241813857

**Published:** 2023-09-08

**Authors:** Masaru Matsuoka, Junko Yamaguchi, Kosaku Kinoshita

**Affiliations:** Division of Emergency and Critical Care Medicine, Department of Acute Medicine, Nihon University School of Medicine, 30-1, Oyaguchi Kami-cho, Itabashi-ku, Tokyo 173-8610, Japan; matsuoka.masaru0416@gmail.com (M.M.); kinoshita.kosaku@nihon-u.ac.jp (K.K.)

**Keywords:** oxidative stress, uric acid, xanthine dehydrogenase, mortality, sepsis

## Abstract

Patient outcomes for severe sepsis and septic shock remain poor. Excessive oxidative stress accelerates organ dysfunction in severe acute illnesses. Uric acid (UA) is the most abundant antioxidant. We hypothesized that UA and related molecules, which play a critical role in antioxidant activity, might be markers of oxidative stress in sepsis. The study aimed to clarify the clinical significance of UA and the relationship between UA, molecules related to UA, and outcomes by measuring blood UA, xanthine dehydrogenase (XDH), and 8-hydroxy-2-deoxyguanosine (8-OHdG) levels over time. Blood UA levels in septic patients were correlated with the SOFA score (ρ = 0.36, *p* < 0.0001) and blood XDH levels (ρ = 0.27, *p* < 0.0001). Blood XDH levels were correlated with the SOFA score (ρ = 0.59, *p* < 0.0001) and blood 8-OHdG levels (ρ = −0.32, *p* < 0.0001). Blood XDH levels were persistently high in fatal cases. Blood XDH level (OR 8.84, 95% CI: 1.42–91.2, *p* = 0.018) was an independent factor of poor outcomes. The cutoff of blood XDH level was 1.38 ng/mL (sensitivity 92.8%, specificity 61.9%), and those 1.38 ng/mL or higher were associated with a significantly reduced survival rate (blood XDH level > 1.38 ng/mL: 23.7%, blood XDH level < 1.38 ng/mL: 96.3%, respectively, *p* = 0.0007). Elevated UA levels due to elevated blood XDH levels in sepsis cases may reduce oxidative stress. Countermeasures against increased oxidative stress in sepsis may provide new therapeutic strategies.

## 1. Introduction

The outcomes for patients with severe sepsis and septic shock remain poor [1]. The sequential organ failure assessment (SOFA) score [2] has been used to identify the severity and degree of organ dysfunction in sepsis. A higher score indicates an increased mortality rate [3]. On the contrary, cases of high SOFA-scoring severe sepsis or septic shock with good prognosis have been reported [4]. Therefore, it is difficult to determine whether these severity scores accurately reflect the pathophysiology. A more sophisticated treatment of sepsis cases with new pathophysiological analyses using adequate markers is needed to improve outcomes. Oxidative stress is defined as an imbalance between oxidants and antioxidants in favor of oxidants, leading to a violation of redox signaling and control and/or molecular damage (Appendix A), which is crucial in the occurrence of organ dysfunction [5]. There are many known markers of oxidative stress and opinions on the validity of oxidative stress markers for the assessment of oxidative stress in various pathological conditions have been presented in recent years [6,7].

However, there are still not enough validated oxidative stress markers known to correlate oxidative stress with the severity and prognosis of sepsis in clinical practice.

Uric acid (UA) is the most abundant antioxidant in vivo. We hypothesized that UA may act as an antioxidant in sepsis and UA levels may serve as markers of oxidative stress in patients with sepsis. Xanthine oxidoreductase (XOR) is an enzyme that catalyzes the synthesis of xanthine and UA from hypoxanthine. XOR exists in two forms: XDH and xanthine oxidase (XO) [8]. XOR is most commonly found in the body as XDH [9,10,11], but XDH and XO are mutually convertible [8]. Since XDH acts as a NADH reductase, reducing nicotinamide adenine dinucleotide (oxidized form; NAD+) to produce NADH and hydrogen ions (H^+^), it does not produce superoxide (O_2_^−^) [12]. In contrast, XO generates O_2_^−^ and hydrogen peroxide (H_2_O_2_) (Appendix A) [6]. The ROS produced as by-products of these enzymatic reactions, namely O_2_^−^ and H_2_O_2_, have been implicated in various oxidative stress pathologies, such as ischemia-reperfusion injury [13].

In addition, 8-OHdG, a by-product of DNA damage, is abundant in all species and is widely used as a biomarker of oxidative stress [14,15]. This study aimed to clarify the clinical significance of UA and molecules related to UA for measurement, analyze the relationship between blood UA levels or blood XDH levels, which promote the production of UA (Appendix A), and blood 8-OHdG level, an indicator of oxidative stress, and determine the outcomes and severity of sepsis patients. 

## 2. Results

### 2.1. Patient Characteristics

During the research period, 75 patients diagnosed with sepsis according to the Sepsis-3 criteria were admitted to the intensive care unit (ICU) at our hospital; 15 of these patients were excluded from this study (Figure 1). None of the patients were prescribed anti-hyperuricemics. Patients were excluded from the study due to lack of consent (n = 11), early death (n = 2), and enrollment in another study (n = 2). Table 1 shows the characteristics of the 60 patients enrolled in this study. The study population had a mean age of 78 ± 12.9 years, including 33 males and 27 females, and 24 patients (40%) diagnosed with septic shock. The SOFA score was 7 (range: 6–10). Sepsis was caused by respiratory infection (n = 34), intra-abdominal infection (n = 3), urinary tract infection (n = 11), or soft tissue infection (n = 3); the focus of infection was unknown in the remaining patients (n = 9). The control group was used for comparison with the experimental groups. Patients in the control group were admitted during the study period in nonseptic conditions.

Patients in the control group (n = 10) were diagnosed with trauma (blunt force, abrasion) (n = 7), loss of consciousness (n = 2), or convulsion (n = 1) at the time of ICU admission. Medical histories included hypertension (n = 1), hypertension and diabetes (n = 1), and renal dysfunction (n = 1); no patients had hyperuricemia or were administered antihyperuricemics. 

### 2.2. Comparison of the Control, Death, and Survival Groups

Table 1 and Table 2 show the comparison of the characteristics and vital signs of patients in the sepsis and control groups. At the time of hospital discharge, none of the control groups showed abnormalities in biochemical data or vital signs. The outcomes at ICU discharge included death in 14 patients (the death group) and survival in 46 patients (the survival group). Table 1 shows the two-group comparison between the death and survival groups. In the death group, the age was significantly higher (*p* = 0.0036), along with the SOFA score at hospital admission (*p* = 0.0041), lactate level (*p* = 0.0024), and blood XDH level (*p* = 0.0004). In contrast, no statistically significant differences were observed between the two groups in blood UA levels (*p* = 0.4471), the amount of UA excreted in the urine (*p* = 0.408), or blood 8-OHdG levels (*p* = 0.0571). 

No statistically significant differences between the two groups were observed in vital signs on hospital admission (Table 2); however, urine volume on day 0 was significantly greater in the survival group (*p* = 0.0001), whereas fluid infusion volumes (*p* = 0.0302) and body fluid balance (*p* = 0.0014) were significantly higher in the death group (Table 3). As a result of investigating whether there was a correlation between blood UA and XDH and the effects of infusion volume and fluid balance, blood UA and blood XDH were not significantly correlated with total infusion volume and fluid balance at 24 h after admission (Appendix A). On the other hand, blood UA and blood XDH showed a weak correlation with fluid balance 72 h after admission. (Blood UA ρ = −0.334, *p* = 0.009, Blood XDH ρ = 0.308, *p* = 0.035) (Appendix A).

Infusion volume: Total infusion volume (mL) 24 or 72 h after hospital admission.

Body fluid balance: Total infusion volume (mL) at 24 or 72 h after admission (urine volume (mL) at 24 or 72 h) + (body fluid loss other than urine). Here, body fluid loss other than urine indicates fluid loss through the gastric tubes or drains is shown.

Percentage change ⊿ (day 1 or day 3 − day 0):

(value on day 1 or 3) − (value on day 0) − (value on day 0)

Non-parametric data are expressed as median values and interquartile ranges.

### 2.3. Correlation of Sepsis Severity with Indicators of Renal Function and Levels of UA, XDH, and 8-OHdG in the Blood

A statistically significant correlation was observed between blood UA levels and SOFA score (ρ = 0.3577, *p* < 0.0001) (Figure 2a), creatinine (Cr) levels (ρ = 0.7623, *p* < 0.0001) (Appendix A, and estimated glomerular filtration rate (eGFR) (ρ = −0.7719, *p* < 0.0001) (Appendix A)). There was a weak correlation (ρ = 0.2717, *p* < 0.0001) (Figure 2b) between blood UA levels and blood XDH levels, and a statistically significant positive correlation was observed between blood XDH levels and the SOFA score (ρ = 0.5852, *p* < 0.0001) (Figure 2c). Blood XDH levels were correlated with Cre (ρ = 0.3817, *p* < 0.0001) (Appendix A) and eGFR (ρ = −0.3841, *p* < 0.0001) (Appendix A). No statistically significant correlation between blood UA levels and blood 8-OHdG levels (ρ = −0.0970, *p* = 0.1592) (Figure 3a) was observed, but a statistically significant negative correlation (ρ = −0.3169, *p* < 0.0001) (Figure 3b) was observed between blood XDH levels and blood 8-OHdG levels.

### 2.4. Change in Blood Levels of UA, XDH, and 8-OHdG over Time

Figure 4a shows changes in blood UA levels. A statistically significant decrease in blood UA levels from day 0 to days 1, 3, 7, and 14 was observed. There was a statistically significant drop on days 3, 7, and 14 compared to that on day 0 (day 3, *p* < 0.0001; day 7, *p* < 0.0001; day 14, *p* < 0.0001) and day 1 (day 3, *p* = 0.0045; day 7, *p* < 0.0001; day 14, *p* = 0.0093). Of the 14 patients in the death group, no change over time on any of the measurement days was observed for blood UA levels (Figure 4a). Figure 4b shows the change in blood XDH levels in the survival group. The results showed a statistically significant decrease in blood XDH levels from day 0 to days 1, 3, 7, and 14. There was a statistically significant drop on days 3, 7, and 14 compared to day 0 (day 3, *p* = 0.0001; day 7, *p* = 0.0001; day 14, *p* = 0.0022). However, no significant changes in blood XDH levels were observed over time in the death group.

In terms of blood 8-OHdG levels, no statistically significant change was observed over time, except on day 1, in the survival group compared to the death group (*p* = 0.001) (Figure 4c).

A comparison of blood UA levels in the control group with those in the survival group showed a statistically significant difference on day 0 (*p* = 0.0037), but no statistically significant differences were observed on days 1, 3, 7, or 14 (Figure 5a). When the levels in the control group were compared with those in the death group, a statistically significant difference was observed on day 0 (*p* = 0.0130), but no statistically significant differences were observed on days 1, 3, 7, or 14 (Figure 5b).

**Figure 5 ijms-24-13857-f005:**
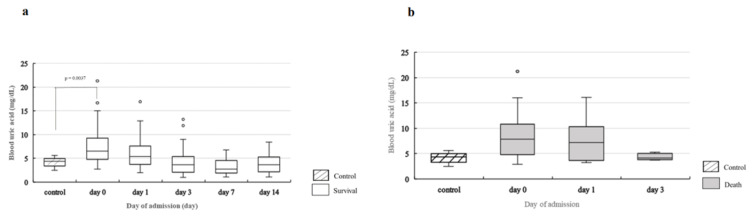
Comparison of blood UA levels in the control group, survival group, and death group. (**a**). Comparison of blood UA levels in the control group and survival group. Statistical analysis is performed using Steel’s test to compare the control and survival groups on each measurement day. Compared with the control group, blood UA levels are significantly higher in the survival group on day 0 (*p* = 0.0037) but no statistically significant differences are observed on the other measurement days. In the survival group, a statistically significant change is observed over time (Figure 4a). (**b**). Comparison of blood UA levels in the control group and death group. Statistical analysis is performed using Steel’s test to compare the control and death groups on each measurement day. Compared with the control group, blood UA levels are significantly higher in the death group on day 0 (*p* = 0.013) but no statistically significant differences are observed on the other measurement days. Comparison of blood XDH levels in the control group with those in the death group showed a statistically significant increase in the death group on days 0 (*p* = 0.0005), 1 (*p* = 0.0001), and 3 (*p* = 0.0012) (Figure 6b). However, no statistically significant differences were observed between the levels in the control group and those in the survival group (Figure 6a). For blood 8-OH-dG levels, no correlation was observed with blood UA levels (ρ = −0.0970, *p* = 0.1592), and the values in the control group and those in the death and survival groups showed no statistically significant differences on any of the measurement days (Figure 7a,b).

**Figure 6 ijms-24-13857-f006:**
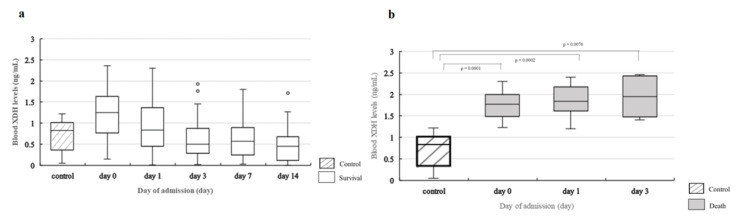
Comparison of blood XDH levels in the control group, survival group, and death group. (**a**). Comparison of blood XDH levels in the control group and survival group. Statistical analysis is performed using Steel’s test to compare the control and survival groups on each measurement day. No significant differences are observed between the control and survival groups on any measurement day. In the survival group, a statistically significant change is observed over time (Figure 4b). (**b**). Comparison of blood XDH levels in the control group and death group. Statistical analysis is performed using Steel’s test to compare the control and death groups on each measurement day. Compared with the control group, blood XDH levels are significantly higher in the death group on days 0 (*p* = 0.0001), 1 (*p* = 0.0002), and 3 (*p* = 0.0076).

**Figure 7 ijms-24-13857-f007:**
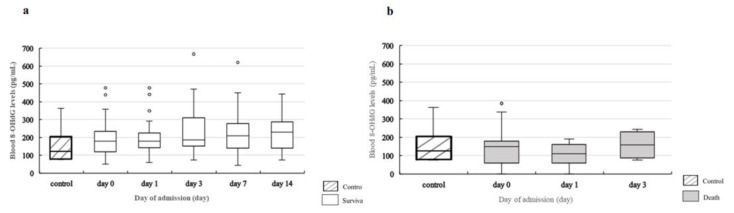
Comparison of blood 8-OHdG levels in the control group, survival group, and death group. (**a**) Comparison of blood 8-OHdG levels between the control group and survival group. Statistical analysis is performed using Steel’s test to compare the control and survival groups on each measurement day. No significant differences are observed between the control and survival groups on any measurement day. (**b**) Comparison of blood 8-OHdG levels between the control group and death group. Statistical analysis is performed using Steel’s test to compare the control and death groups on each measurement day. No statistically significant differences are observed between the control and death groups on any measurement day.

### 2.5. Relationship with Factors Affecting Outcomes

A univariate logistic regression analysis was performed using data from the time of hospital admission to investigate the factors having an impact on outcomes. A statistically significant relationship with death was observed for the following factors: age (OR 1.12, 95% CI: 1.02–1.22, *p* = 0.002), blood XDH levels (OR 14.25, 95% CI: 2.51–80.79, *p* = 0.0002), SOFA at hospital admission (OR 1.46, 95% CI: 1.13–1.97, *p* = 0.0025), Cr (OR 1.413, 95% CI: 1.026–2.018, *p* = 0.035), and lactate levels (OR 1.367, 95% CI: 1.120–1.668, *p* = 0.0003).

We used age, SOFA score at admission, lactate level at admission, and blood XDH levels as covariates and performed a multivariate logistic regression analysis. Factors included in the SOFA score were not fed into the multivariate logistic regression analysis, and blood UA levels demonstrated a strong correlation with Cre (ρ = 0.76, *p* < 0.0001), so they were excluded as covariates. No statistically significant relationship between outcomes for SOFA score and hospital admission (OR 1.169, 95% CI: 0.819–1.704, *p* = 0.384), age (OR 1.089, 95% CI: 0.995–1.229, *p* = 0.068), or lactate levels (OR 1.223, 95% CI: 0.993–1.602, *p* = 0.059) was observed, but a relationship with outcomes (death) for blood XDH levels (OR 8.839, 95% CI: 1.417–91.21, *p* = 0.018 (Table 4)) was found. Multicollinearity is assessed using variance inflation factors (VIF) [16]. The potential multicollinearity was found to be present in age, SOFA score, lactate, and blood XDH levels (VIF = 1.121, 1.242, 1.310, and 1.191, respectively). When comparing the ICU length of stay for all patients, the survival group was significantly longer than the death group. (8 days (3–73 days) vs. 2 days (1–22 days), respectively, *p* < 0.001). Blood UA and XDH levels on admission were not correlated with ICU length of stay (Appendix A).

The receiver operating curve (ROC), which was plotted for XDH levels at hospital admission, is shown in Figure 8. The area under the curve (AUC) ROC score was 0.81 for blood XDH levels (*p* = 0.0002). When the cutoff value for blood XDH was derived, the sensitivity and specificity were 92.8% and 61.9%, respectively, at 1.38 ng/mL. Using this value, we divided the sepsis patients into two groups (blood XDH levels of >1.38 ng/mL and <1.38 ng/mL) and investigated survival rates at the time of ICU discharge. In the blood XDH > 1.38 ng/mL group, the log-rank test showed a statistically significant reduction in survival rate (*p* = 0.0007). The survival rates were 23.7% for the >1.38 ng/mL group and 96.3% for the <1.38 ng/mL group (Figure 9). 

For the validation cohort to predict the prognosis of blood XDH levels, bootstrapped c statistics [17] and calibration curves were used to assess external model discrimination and fit. We generated 1000 ROC (receiver operating characteristic) curves using the bootstrap method and calculated 95% confidence intervals for the AUC (area under the curve). As a result, the AUC (95% confidence intervals) is 0.816 (0.67–0.92), and the *p*-value in the Hosmer–Lemeshow good fit test is 0.769.

We divided the groups into two groups: the training group and the test group [18]. We split the data 4:1 or 2:1 (random assignment) and obtained AUC values and 95% confidence intervals for the training and test groups, and the Hosmer–Lemeshow test was performed to validate the optimization of the model obtained by multivariate logistic regression analysis of the training data [19,20]. In both distribution method studies, the 95% confidence intervals of the AUC decreased in the training cohort and increased in the test cohort compared to the original predictive outcome of blood XDH levels. The AUC [95% confidence intervals] is 0.816 [0.67–0.92] (Appendix A). The result of the calibration curve of the blood XDH prediction nomograms does not always indicate good agreement between prediction and observation in the training and test groups (Appendix A).

## 3. Discussion

In this study, we observed a statistically significant reduction in survival rates at ICU discharge in patients with high blood XDH levels upon hospital admission. We also observed a positive correlation with the SOFA score, an indicator of disease severity in patients with sepsis, and a clear relationship between XDH levels and death from sepsis, with patients who died exhibiting persistently high XDH levels. The Third International Consensus Definitions for Sepsis and Septic Shock (Sepsis-3) define sepsis as life-threatening organ dysfunction caused by an abnormal host response to infection [21]. The SOFA score is used in actual clinical practice as a diagnostic criterion, severity assessment, and predictor of outcome for sepsis by assessing organ failure and determining severity with an overall score. Oxidative stress due to excessive ROS production may play an important role in the progression of organ failure in tissue inflammation, ischemia and hypoxia caused by infection, burns, trauma and ARDS, and ischemia-reperfusion in cerebral infarction, acute myocardial infarction, intestinal ischemia, and organ transplantation [22].

Sepsis is a complex condition that combines inflammation, hypoxia, ischemia, and ischemia-reperfusion as described above, suggesting that oxidative stress may contribute to organ dysfunction.

Although sepsis is one of the most complex pathologies in which the inflammatory response to pathogens becomes unbalanced and leads to organ failure [23], endothelial cell dysfunction and microvascular dysfunction are the major causes of tissue hypoperfusion and multiorgan failure that occur in sepsis. Under normal conditions, the cellular endothelium maintains an appropriate oxidative-antioxidative balance and appropriate redox reactions (signaling), and in this response, reactive oxygen species are continuously supplied by endothelial metabolism [24]. However, in sepsis, oxidative stress is generated by a variety of mechanisms, and the endothelium itself is a target of oxidative stress. Many functions of the endothelium are altered, resulting in phenotypes such as abnormal vascular tone, proinflammatory, procoagulant, glycocalyx disruption, and leukocyte adhesion (sepsis-induced endotheliopathy), which in turn leads to further oxidative stress, resulting in a vicious cycle that further disrupts the multiple functions of the microvasculature, which in turn leads to further oxidative stress. This ultimately promotes cell death and leads to organ failure. During sepsis, PAMPs and DAMPs accumulate antioxidants from intracellular enzymes such as NADPH and XO, uncoupling mitochondria and eNOS in the cell endothelium. During ischemia, there is an accumulation of purines such as hypoxanthine and xanthine, which are substrates of XO, and a depletion of intracellular ATP, but when oxygen delivery is restored by reperfusion, hypoxanthine, and xanthine are converted by XO to UA, and superoxide is produced [25]. The results of this study indicate blood XDH levels correlate with the severity of sepsis as indicated by the SOFA score and may be a prognostic factor for poor prognosis.

XDH also functions as an NADH oxidase [26] that oxidizes NADH and produces O_2_^−^ and XDH can produce four times the amount of ROS compared with XO [27]. As a result, both XDH and XO produce ROS in the body and may cause tissue injury. However, in the sepsis patients in this study, blood XDH levels were positively correlated with blood UA levels but negatively correlated with blood 8-OHdG levels. Therefore, XDH may not function to produce ROS but instead produce the antioxidant UA, giving rise to the possibility that blood XDH may contribute to reducing oxidative stress in the body.

Blood XDH levels were persistently high in the death group. XOR is released during the physiological turnover of cells and from cells in a diseased state [28] and is most commonly distributed in epithelial cells in the small intestine and liver [29]. These organs release XOR into the blood from cells injured by ischemia-reperfusion injury following liver transplantation [30,31,32], extensive intestinal ischemia [33,34], and hypoxia [35], which are pathologies similar to those of total body injury. However, ROS produced from XOR inside cells triggers cell injury [36], and blood XOR can spread to all organs in the body through the blood, causing tissue injury in remote organs, including the lungs [37] or kidneys [38], and triggering ARDS, or multiple organ failure [39]. The mechanism is thought to involve binding to glycolipids present on the surface of vascular endothelial cells in remote organs [40] and cell uptake, which triggers injury due to reactive oxygen originating from blood XOR [41] or expression of cell adhesion molecules (CAMs) and neutrophil stimulation promoting extravascular infiltration [42]. This suggests that XOR is not only released into the blood after cell injury but is also involved in the progression of organ dysfunction. Persistently high blood XOR levels reflect not only organ damage but may also be an actual cause of organ damage. This study focused on UA, and logistic regression analysis showed that blood UA level was not a prognostic factor but identified only blood XDH level as an independent prognostic factor. Uric acid and XO activity are two-faced Janus in the biochemistry of oxidative stress; some of the papers that have been reported have produced conflicting results in animal models of sepsis, and there are not as many reports on the relationship between UA levels and outcome in human sepsis cases. Some of the papers that have been reported have produced conflicting results. Many reports that hyperuricemia is associated with sepsis [43,44,45,46], sepsis-induced ARDS [47], AKI prognosis [48], prolonged hospital stay [43], or that uric acid increases over time [49]. On the contrary, some reports suggest that hypouricemia may be a poor prognostic factor for sepsis or sepsis-associated ARDS [50,51]. We believe that how this uric acid paradox operates in the septic state remains unexplained.

Although uric acid levels remain high in humans because urate oxidase is not activated and uric acid is the end product of purine catabolism, uric acid is a potent antioxidant, and some have suggested that the acquisition of antioxidant capacity due to defective expression of urate oxidase protein may have favored long-term survival in humans [52]. In contrast, the papers [53] reported that hyperuricemia is a risk factor for poor outcomes in the general population and patients with chronic heart failure.

An explanation for the “double-faced Janus” of uric acid is provided by Sautin et al. [54] in a review article showing that uric acid is mainly an antioxidant in hydrophilic environments such as plasma and hydrophobic environments such as intracellular hydrophilic environments such as plasma, and a pro-oxidant in hydrophobic environments such as the intracellular environment.

It has been reported to enhance the oxidation of LDL and liposomes by the oxidant peroxynitrite at high uric acid concentrations above physiological levels [55]. The same authors [56] also found that uric acid in mature adipocytes stimulated NADPH oxidase activity, increased ROS, decreased NO bioavailability, and increased protein nitrosylation, suggesting that hyperuricemia increases redox-dependent signaling and oxidative stress primarily in adipocytes.

The fact that uric acid is a potent scavenger of singlet oxygen, peroxyl radicals, and hydroxyl radicals in hydrophilic environments such as plasma, but is unable to supplement lipid-soluble radicals and prevent radical propagation in lipid membranes, suggests that oxidative stress and protein and lipid oxidation modification reactions are common to all pathologies, but may explain the relationship between all diseases and outcomes, as well as the different results regarding the cardiovascular risk of hyperuricemia in obesity.

At this point, the clinical significance of hyperuricemia and blood XDH levels in sepsis is discussed in terms of only limited results of the study, however, the blood XDH level may reflect the severity of sepsis pathology more sensitively than the blood UA concentration.

Therefore, for patients with sepsis, it is possible that an increase in blood XDH levels could be used as a pathophysiological biomarker to predict sepsis outcomes. The levels of enzymes LDH, AST, and ALT increase as they are released into the blood from cells that are breaking down, and we believe that blood XDH levels rise through a similar mechanism, that is, due to organ injury accompanying sepsis. However, we did not investigate whether XDH stored within the liver and small intestine [29] was released directly into the blood.

Elevated XO/XOR ratios have been reported in patients with chronic kidney disease [57,58,59]. XOR inhibitors have been highlighted as having therapeutic potential for these chronic conditions as they continuously suppress ROS production by the biosynthetic pathway converting xanthine to UA [25]. Reports on the potential effects of the XOR inhibitor, allopurinol, in reducing cardiovascular events in patients with hypertension and reduced kidney function [58] or an increase in systemic XO activity in patients with coronary artery disease [60] support the idea that XOR inhibitors may also reduce oxidative stress and play an important therapeutic role [61]. However, there are numerous reports on the antioxidant actions of UA, and UA may improve functional outcomes, based on reports on the involvement of gout caused by hyperuricemia in improvements in patients with Parkinson’s disease [62], Alzheimer’s disease [63], and vascular/non-vascular dementia [64]. The administration of UA during the acute phase of ischemic stroke may be beneficial [65]. While there are expectations that XOR inhibitors may be effective treatments for chronic diseases such as chronic renal failure or cardiovascular disease, a consensus has yet to be reached regarding the effects of XOR inhibitors against the acute onset of sepsis [66]. There are still few reports in the literature supporting a role for XO in the pathogenesis of human sepsis [67,68,69] We believe that there is a need to analyze uric acid and XDH levels in different physicochemical situations and at different tissue and organ levels to be needed.

The literature includes reports of potential treatments using various antioxidants [70,71,72]. Recent reports have discussed the potential for sepsis patients to be treated with a combination of hydrocortisone, vitamin C, and thiamine [4,72], and we expect future research into anti-inflammatory and antioxidant therapies for sepsis based on a report on the relationship between the inflammatory response and oxidative stress [73]. Blood UA levels are generally thought to exhibit a negative correlation with GFR [74], dehydration [75], renal function, and a decline in the amount of UA excreted in the urine [76]. However, blood XDH levels increased in patients with severe sepsis, and there was a positive correlation with Cr and a negative correlation with eGFR.

Differences in urine volume, infused fluid volume, and body fluid balance were observed between patients with sepsis in the death and survival groups. However, there were no differences in the changes in blood UA levels between the two groups during the research period. In the results of the study, the correlation between blood UA and blood XDH and the effect of transfusion volume correlated with fluid balance after 3 days of hospitalization (Appendix A). These results suggest that the increase in blood UA and blood XDH levels due to sepsis may be related to decreased fluid volume rather than renal function.

This study had several limitations. Univariate analysis showed that UA was not an independent prognostic factor; however, we need to consider confounding factors and increase the number of patients. Multivariate analysis showed that blood XDH level was an independent prognostic factor for outcome, but the possibility of weak multicollinearity was observed for each factor. The effect was not considered severe, the number of only sixty cases in our study is small, and the confounding effect should be considered in many future cases of sepsis.

Although the literature includes reports on the measurement of urinary 8-OHdG as a marker of oxidative stress [77], we evaluated blood 8-OHdG levels in this study because oliguria was common in patients in the death group. The results did not show a direct correlation between increased blood UA and decreased blood 8-OHdG levels. Blood 8-OHdG levels reflect the progression of oxidative stress within the body, and levels can change not only when there is a disease but also because of lifestyle factors such as aging, exercise, diet, smoking, and sleep [78,79,80]. Blood UA acts as an antioxidant, but multiple mechanisms within the body protect against oxidation, so it may be that these other mechanisms played a role in the antioxidant effects, resulting in no direct correlation being observed between blood UA levels and blood 8-OHdG levels. 

ROS oxidizes lipids, proteins [5], and DNA, thus damaging biological functions. This study did not measure these oxidative stress markers; therefore, future research could adopt a multilateral approach by measuring other oxidative stress markers. Additionally, blood XO levels were not measured; therefore, the relationship between blood UA and 8-OHdG levels was not investigated.

For XDH produced in vivo, the extent to which XDH is converted into XO and whether there is any impact on oxidative stress should be determined. One study reported that beta-lactam compounds blocked XOR [81]; however, this study did not investigate the effects of antibacterial agents on blood XDH.

The results demonstrated a positive correlation between blood XDH levels and SOFA scores, which are related to sepsis severity. The XOR gene is activated by various factors, including interferon-gamma (IFN-ɣ), interleukins (IL) 1 and 6, glucocorticoids, hypoxia, and lipopolysaccharides [82]. We did not measure cytokines and other mediators in the blood and did not investigate their relationships with blood XDH levels. There is a strong correlation between increased XOR activity in the blood and NF-κB activity and IL-6 levels [83], and another report shows inflammatory mediators and cytokines produced during the host response to infection increased XOR expression along with enhanced neutrophil activation [84]; however, we did not investigate the possibility that multiple factors may be involved in the increased blood XDH levels seen with sepsis, including hypercytokinemia accompanying an excessive systemic inflammatory response caused by infection, ROS produced from XO acting on the vascular endothelial cells causing coagulation disorders [85], or the impacts of specific pathogens [86,87].

## 4. Materials and Methods

### 4.1. Study Subjects

This was a single-center, prospective observational study conducted at Nihon University Itabashi Hospital (study no. RK-170912-08) between February 2018 and May 2019. The study enrolled patients meeting the diagnostic criteria and definition of sepsis according to the Surviving Sepsis Campaign Guidelines (SSCG) 2016 (Sepsis-3) [21] from among consecutive patients transferred to Itabashi Hospital’s Emergency and Critical Care Center and admitted to the ICU. The exclusion criteria were: (1) age < 20 years, (2) in-hospital sepsis, and (3) transfer from another hospital with therapeutic intervention started before admission to our institution. For the control group, we enrolled patients aged > 20 years who were diagnosed with sepsis based on the above criteria among patients transferred and admitted to the ICU between February 2018 and May 2019 (Figure 1).

### 4.2. Measurement of Samples

The patient’s vital signs were recorded on the day of admission (day 0) and days 1, 3, 7, and 14 after admission; blood biochemistry test data were used to assess blood UA levels and blood XDH levels, related to the production of UA (Appendix A) and blood 8-OHdG levels as an indicator of oxidative stress. Blood UA value is influenced by dehydration (body fluid volume [extracellular fluid volume]) or urinary excretion of UA. Total infusion volume, body fluid balance, and urinary excretion of UA were calculated (see below). XOR is an enzyme that catalyzes the synthesis of xanthine and UA from hypoxanthine. XOR exists in two forms: XDH and XO [6]. For measurements of blood XDH levels and blood 8-OHdG levels, 5 mL blood samples were collected simultaneously and immediately centrifuged to separate the serum, which was then frozen at −80 °C until measurement of blood levels of both enzymes. An enzyme-linked immunosorbent assay (ELISA) kit (Cloud-Clone Corp. Houston, TX 77084, USA) [88] and an ELISA kit for humans (Cloud-Clone Corp. Katy, TX 77494, USA) [89] were used to measure blood XDH and 8-OHdG levels. The ELISA kit used in this study is usually less sensitive to the components present in the sample (serum) and to dilution, resulting in less variation in the assay data. 8-OHdG is a very stable byproduct of DNA damage and is used as a marker of oxidative stress [14]. Since this study used human serum and not cell and tissue samples, DNA extraction was not required because the samples contained high levels of 8-OHdG.

All samples were subjected to measurement twice, and the mean value was used as the measured result. 

The control group consisted of ill patients who were admitted to the ICU during the study period but were not diagnosed with sepsis. In the control group, vital signs, blood biochemistry, and urine tests were performed prior to discharge. The time before discharge was considered unaffected by each disease. To clarify the clinical significance of these markers, we compared the study and control groups.

The SOFA score was calculated at ICU admission as an indicator of sepsis severity. The SOFA score evaluates each of organs: CNS function (Glasgow Coma Scale or GCS), cardiovascular abnormalities (blood pressure, vasopressor levels), coagulation disorders (platelet count), hepatic dysfunction (total bilirubin), and renal dysfunction (Cr).

The outcomes were evaluated as survival or death at the time of discharge.

To clarify the relationship between various blood and urine sample measurements (blood UA level, blood XDH level, blood 8-OHdG, etc.), the survival and death groups, the control and survival groups, and the control and death groups were compared using statistical analysis methods.

Urinary excretion of UA (mg/g/Cr)

UA is excreted by the kidneys; therefore, the amount of UA excreted in urine affects the level of UA in the blood [41]. Hyperuricemia is an indicator of dehydration [40] and may be affected by the hydration levels in the body. We also thought that changes in body hydration levels due to fluid infusion may affect blood UA levels. Therefore, we defined and calculated the total infusion amounts and body fluid balance to determine whether body hydration levels after hospital admission affected blood UA levels.

UA and Cr were measured in spot urine samples, and the amount of UA excreted in the urine was estimated using the following formula: 

Urinary excretion of UA (mg/g/Cr) = Urinary UA level (mg/dL)/urinary Cr level (mg/dL)

To compare the changes in blood UA and XDH levels from day 0, the percentage change (shown as ⊿) on days 1 and 3 was derived using the following formula:

Percentage change ⊿ (day 1 or day 3 − day 0) = 

(value on day 1 or 3) − (value on day 0)/(value on day 0)

Infusion volume

Total infusion volume (mL) at 24 or 72 h after hospital admission

Body fluid balance

Total infusion volume (mL) at 24 or 72 h after admission − (urine volume (mL) at 24 or 72 h) + (body fluid loss other than urine) 

Urinary excretion of UA was measured on days 0, 1, 3, and 7. Patients unable to provide urine samples, for example, due to oliguria accompanying acute renal impairment, were excluded from the study.

### 4.3. Statistical Methods

Statistical analyses were performed using the JMP pro 13.2.1 statistical software package (SAS Institute, Cary, NC, USA), and the C-index and calibration curve were built by being computed on R version 4.3.1 (R Foundation for Statistical Computing) with the “rms” package and “calibration curve” package. The concordance index (C-index) was computed on R with the “rms” package and the “Hmisc” package [90]. The Shapiro–Wilk test was used to evaluate data normality, with the significance level set at <5%. Data are expressed as integers (%) for discrete variables. For continuous variables, data in a normal distribution were expressed as mean ± SD, while data in a non-normal distribution were expressed as median values and the corresponding interquartile ranges. The chi-square (χ^2^) test was used to compare data categories. Comparisons between the two groups were performed using the Student’s *t*-test for parametric data and the Wilcoxon rank sum test for non-parametric data. Two-tailed tests were performed on both data types with a 5% hazard ratio. For multiple comparisons, the Kruskal–Wallis test was used for non-parametric data in unmatched groups, and the Friedman test was used for non-parametric data in matched groups. The Steel–Dwass test was used for posthoc tests. The Steel test was used to compare the control and experimental groups. The Spearman’s rank correlation test was used to test for correlations. 

A univariate logistic regression analysis was performed for all explanatory variables to investigate their impact on outcomes at ICU discharge. A ROC curve was plotted for the explanatory variables of UA and UA synthase, and the cutoff value was determined using the Youden Index. The data were divided into two groups based on cutoff values, and the Kaplan-Meier method was used to calculate survival rates at the time of ICU discharge, which were compared using the log-rank test. After adjusting for patient characteristics at the time of hospital admission and disease severity, logistic regression analysis was performed to investigate whether UA and UA synthase were independent predictors of sepsis outcomes. A univariate logistic regression analysis was performed with all data at the time of hospital admission as independent variables. In sepsis, higher disease severity scores, older age [91], and elevated lactate levels [92] are known independent prognostic factors. Consideration was given to not including strongly correlated variables arising from multiple collinearity or linearly dependent variables (Cr, bilirubin, platelet count, PaO_2_/FiO_2_ ratio, GCS, mean blood pressure) included within the scores, as seen with SOFA scores and Cr. Multiple logistic regression analysis was performed, and the *p*-value, odds ratio, and 95% confidence intervals (CI) were calculated. To perform the validation cohort of the obtained prognostic value, the bootstrap method was used for internal validation [17]. The 95% CI and AUCs were obtained by creating ROC curves with 1000 repetitions. The Hosmer–Lemeshow test was conducted to validate the optimization of the model obtained by multivariate logistic regression analysis of training data [19,20] The C-index was calculated, and calibration plots were created to check whether the predicted results matched the actual distribution of the data [18]. The significance level was set at *p* < 0.05.

## 5. Conclusions

This study showed a statistically significant decrease in survival rates upon ICU discharge in patients with high blood XDH levels upon hospital admission. A positive correlation between blood XDH levels and the SOFA score, an indicator of sepsis severity, was found. Patients who died had persistently high blood XDH levels, which were related to death from sepsis. Regarding blood UA levels, there may be a relationship between increased blood XDH levels due to sepsis and increased blood UA levels caused by the action of this enzyme, as well as effects on renal function and reduced body fluid volumes caused by dehydration. In addition, a negative correlation was observed between blood XDH levels and blood 8-OHdG levels, suggesting that in patients with sepsis, UA increases due to the rise in blood XDH, and this may play a role in reducing oxidative stress within the body. In conclusion, in addition to targeting excessive inflammatory responses in sepsis, we believe that measures to address concomitant oxidative stress are new therapeutic targets.

## Figures and Tables

**Figure 1 ijms-24-13857-f001:**
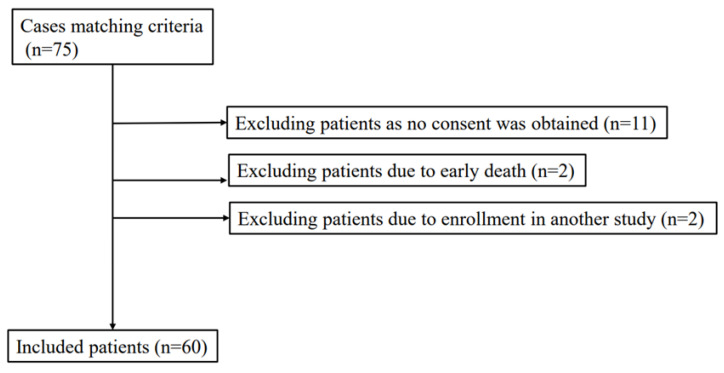
Consort diagram showing the inclusion process. During the research period, 75 patients diagnosed with sepsis according to the Sepsis-3 criteria were admitted to the ICU at our hospital; 15 of these patients were excluded from this study. None of the patients was prescribed antihyperuricemics. Patients were excluded from the study due to lack of consent (n = 11), early death (n = 2), or enrollment in another study (n = 2).

**Figure 2 ijms-24-13857-f002:**
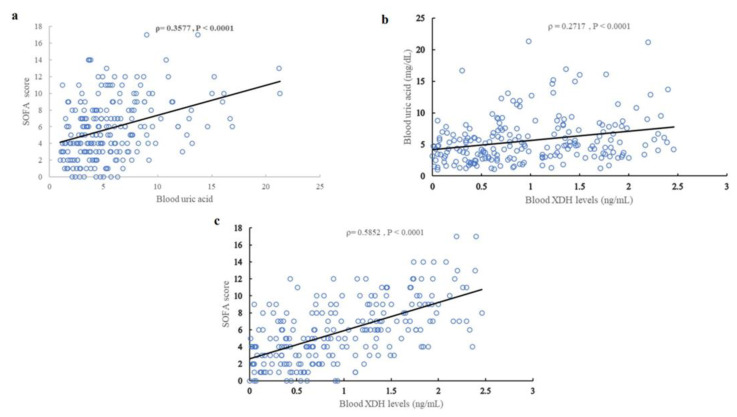
Correlation between sepsis severity, indicators of renal function, and levels of UA. (**a**) Blood UA levels and SOFA scores in sepsis patients. (**b**) Blood XDH levels and blood UA levels in sepsis patients. (**c**) Blood XDH levels and SOFA scores in sepsis patients. Blue circles shows the actual values for each patient. Black line shows regression line.

**Figure 3 ijms-24-13857-f003:**
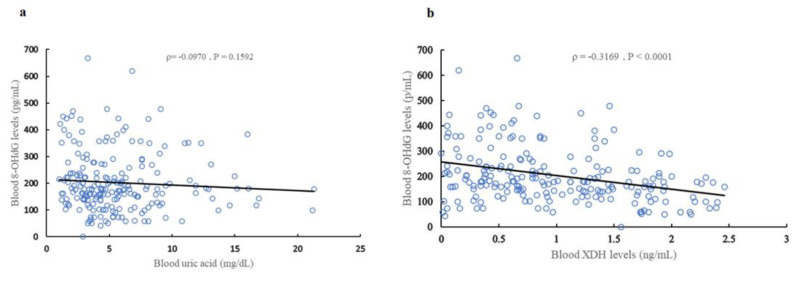
Correlation between blood 8-OHdG levels and XDH, and blood UA levels (**a**) Blood UA levels and blood 8-OHdG levels in sepsis patients. (**b**) Blood XDH levels and blood 8-OHdG levels in sepsis patients. Blue circles shows the actual values for each patient. Black line shows regression line.

**Figure 4 ijms-24-13857-f004:**
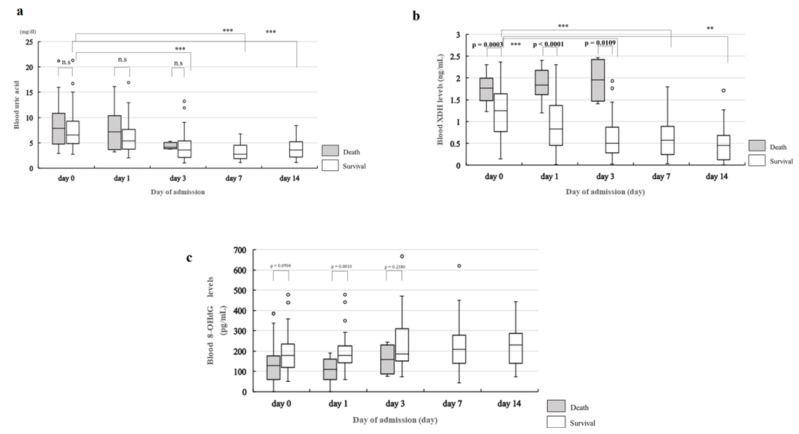
Change in blood UA, blood XDH, and blood 8-OHdG levels over time. (**a**) Change in blood UA levels over time. Statistical analysis is performed using the Friedman test and Steel–Dwass test for group comparisons between three patients in the death group and the five patients in the survival group. No change over time is observed in the death group (*p* = 0.7364); however, a statistically significant change over time is observed in the survival group (*p* < 0.0001). In the survival group, compared with day 0, there is a significant decrease on day 3 (*p* < 0.0001), day 7 (*p* < 0.0001), and day 14 in the death group. Only two patients survived beyond day 7, so we used data up to day 3 to investigate changes over time (day 0: n = 14; day 1: n = 13; day 3: n = 5). (**b**) Change in blood XDH levels over time. Statistical analysis is performed using the Friedman test and Steel–Dwass test for group comparisons between the three patients in the death group and the five patients in the survival group. No change over time is observed in the death group (*p* = 0.7023); however, a statistically significant change over time is observed in the survival group (*p* < 0.0001). In the survival group, compared with day 0 there is a significant decrease on days 3 (*p* = 0.0001), 7 (*p* = 0.0001), and 14 (*p* = 0.0022). Comparison between the death group and survival group on each measurement day shows significantly higher blood XDH levels in the death group on days 0 (*p* = 0.0003), 1 (*p* < 0.0001), and 3 (*p* = 0.0109). (**c**) Change in blood 8-OHdG levels over time. Statistical analysis is performed using the Friedman test and Steel–Dwass test for group comparisons between the three patients in the death group and the five patients in the survival group. No change over time is observed in the death group (*p* = 0.7338) or the survival group (*p* = 0.2968). Comparison between the death group and survival group on each measurement day shows significantly lower blood 8-OHdG levels in the death group on day 0 (*p* = 0.001). ** means *p* < 0.001, *** means *p* < 0.0001.

**Figure 8 ijms-24-13857-f008:**
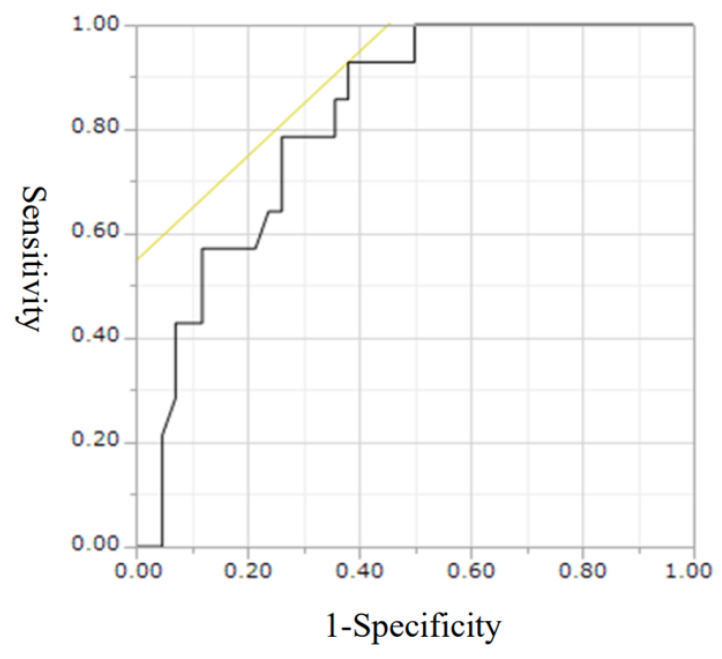
ROC plot of blood XDH levels at hospital admission (day 0). Area under the curve (AUC): 0.8163. Cut-off value: sensitivity 92.8% and specificity 61.9% at 1.38 ng/mL. Yellow line; A 45 degree straight line tangent to the ROC curve, an indicator of the optimal cut-off.

**Figure 9 ijms-24-13857-f009:**
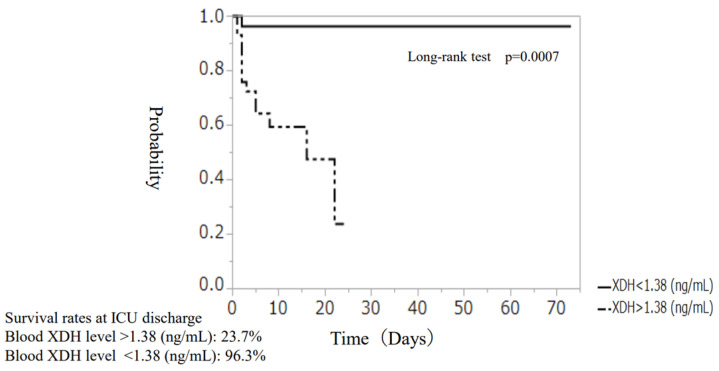
Survival curve at ICU discharge (investigated by dividing patients into two groups based on blood XDH levels at admission). Survival rates at ICU discharge. Blood XDH level > 1.38 (ng/mL): 23.7% Blood XDH level < 1.38 (ng/mL): 96.3%.

**Table 1 ijms-24-13857-t001:** Comparison of characteristics of the sepsis and control groups.

Background	Total (n = 60)	Death (n = 14)	Survival (n = 46)	Control (n = 10)	P_1_	P_2_
Age	78.4 ± 12.9	86.1 ± 8.3	76.1 ± 13.2	53.0 ± 21.5	0.0036	0.0007
Gender (female/male)	27/33	4/10	23/23	6/4	0.1582	N.A
BMI (kg/m^2^)	20.3 ± 4.6	20.1 ± 2.9	20.3 ± 5.1	22.7 ± 4.1	0.8203	0.1224
Septic shock, n (%)	24 (40)	11 (78)	13 (28)	N.A	0.0008	N.A
SOFA score	7 (6–10)	9.5 (7.75–12)	7 (5–9.25)	2 (0.75–4)	0.0041	<0.0001
WBC (×10^3^/μL)	10.5 ± 5.5	10.3 ± 6.2	10.5 ± 5.3	6.5 ± 2.1	0.7664	0.0271
Platelet (×10^4^/μL)	203.5 (129.3–297.3)	163 (91–306)	210.5 (134–289.5)	233.7 (183.3–286.3)	0.2082	0.5346
CRP (mg/dL)	7.7 (1.7–17.5)	10.7 (1.1–27.0)	7.1 (1.7–15.2)	0.14 (0.18–0.1)	0.3966	<0.0001
Cr (mg/dL)	1.44 (0.98–2.14)	1.76 (5.29–1.11)	1.33 (0.85–1.90)	0.66 (0.58–0.7)	0.0836	0.0001
eGFR (mL/min/1.73 m^2^)	35.2 (25.5–52.7)	30.4 (16.3–43.55)	37.6 (27.0–59.4)	84.4 (72.7–94.6)	0.1137	0.0003
T.bilirubin (mg/dL)	0.64 (0.42–1.07)	0.64 (0.36–1.60)	0.62 (0.42–1.03)	0.73 (0.49–1.09)	0.9721	0.8338
LDH (U/L)	298 (224.6– 386.8)	323 (256.3–522.5)	289 (221.3–373.8)	177 (141.8–214.0)	0.2312	0.0002
AST (U/L)	46.5 (24.25–70.75)	44 (22–114.75)	46.5 (24.75–70)	18.5 (14.5–21.25)	0.6811	0.0001
ALT (U/L)	24.5 (14.25–50.5)	25.5 (14–91.75)	24.5 (14.75–43)	12 (9–16.5)	0.8818	0.0025
Blood.UA (mg/dL)	7.0 (4.8–9.4)	7.9 (4.8–10.9)	6.6 (4.8–9.3)	4.4 (3.4–5.0)	0.447	0.0007
Urinary.UA Excretion (mg/g·Cr)	0.38 (0.21–0.60)	0.37 (0.06–0.51)	0.38 (0.21–0.64)	N.A	0.408	N.A
Lactate (mmoL/L)	3.3 (2.0–7.1)	7.5 (3–11.5)	2.8 (1.9–5.2)	1.1 (0.9–2.2)	0.0024	0.0003
Blood XDH (ng/mL)	1.31 ± 0.56	1.76 ± 0.32	1.16 ± 0.55	0.70 ± 0.38	0.0004	0.0023
Blood 8-OHdG (pg/mL)	171.5 (101–222.25)	148.0 (58–178.5)	179.5 (118–234)	124.5 (77.5–205.5)	0.0571	0.3663
MAP (mmHg)	87 (67.25–107.25)	90.5 (65.3–114.5)	85 (68.8–105.8)	112 (88.8–126.8)	0.9721	0.006
HR (/min)	120.5 (99.25–143.5)	110 (99.3–141.8)	121.5 (98–144.3)	80.1 ± 16.4	0.5065	0.0004
RR (/min)	28 (22–32)	29 (20.3–30.5)	27.5 (22–37.3)	20.7 ± 5.14	0.9094	0.0363
BT (°C)	37.2 (35.9–38.9)	37.7 (38.9–35.8)	36.9 (35.9–38.9)	36.0 (34.7–36.7)	0.875	0.0167
Urine volume (mL/day)	950 (354–2080)	235 (122.3–795)	1305 (651.3–2135)	N.A	0.0001	N.A
Infusion volume (mL/day)	5151 (3855–6687)	5931 (4882–10144)	5000 (3579–5975)	N.A	0.0302	N.A
Water balance (mL/day)	3695.5 (2097–5387)	5336.5 (3646–9246)	2985 (1918–4375)	N.A	0.0014	N.A

P1: Death group vs. survival group; P2: All patients with sepsis vs. the control group; NA: not available. Parametric data are expressed as mean ± SD and non-parametric data are expressed as median values and interquartile ranges. Comparisons between the two groups are performed using the Student’s *t*-test for parametric data and the Wilcoxon rank sum test for non-parametric data. The significance level is set at *p* < 0.05.

**Table 2 ijms-24-13857-t002:** Comparison of vital signs at hospital admission.

At Admission	All	Death	Survival	Control	P3	P4
MeanBP (mmHg)	87 (67.3–107.3)	90.5 (65.3–114.5)	85 (68.8–105.8)	112 (88.8–126.8)	0.9721	0.0363
HR (/min)	120.5 ± 34.8	115.9 ± 24.9	121.9 ± 37.4	80.1 ± 16.4	0.5065	0.0004
RR (/min)	27.26 ± 8.33	27.6 ± 7.41	27.15 ± 8.67	20.7 ± 5.14	0.9094	0.0060
BT (°C)	37.2 (35.9–38.9)	37.6 (35.8–38.9)	36.9 (35.9–38.9)	35.95 (34.7–36.7)	0.875	0.00167

P3: Death group vs. survival group; P4: All sepsis patients vs. control group; NA: not available; Parametric data are expressed as mean ± SD and non-parametric data are expressed as median values and interquartile ranges. Comparisons between the two groups are performed using the Student’s *t*-test for parametric data and the Wilcoxon rank sum test for non-parametric data. The significance level is set at *p* < 0.05.

**Table 3 ijms-24-13857-t003:** Comparison of factors affecting body fluid amounts after hospital admission in sepsis patients (urine volume, fluid infusion volume, and body fluid balance) and percentage change ⊿.

At 24 h after Admission	All	Death	Survival	*p*-Value
Urine volume (mL)	950 (354–2080)	235 (122–795)	1305 (651–2135)	0.0001
Total infusion volume (mL)	5151 (3855–6686)	5930 (4882–10144)	5000 (3578–5975)	0.0302
Body fluid balance (mL)	3695 (2097–5387)	5336 (3645–9246)	2985 (1918–4374)	0.0014
UA⊿ (day1–day0)		−4.93 (−29.06–12.68)	−15.87 (−29.06–−3.50)	0.2068
XDH⊿ (day1–day0)		−3.33 (−16.03–27.48)	−14.38 (−60.47–13.44)	0.1169
**At 72 h after admission**				
Urine volume (mL)	4375 (1573–6247)	421 (174–1275)	4987 (3751–7463)	<0.0001
Total infusion volume (mL)	9824 (6918–12256)	10209 (8324–4105)	9747 (7954–12,196)	0.4419
Body fluid balance (mL)	4848 (2537–7402)	9400 (5418–13226)	4066 (1648–5668)	<0.0001
UA⊿ (day3–day0)		−19.60 (−70.55–−32.45)	−40.31 (−63.85–−26.14)	0.3321
XDH⊿ (day3–day0)		−0.70 (−14.10–47.92)	−50.72 (−74.91–−19.51)	0.0056

Comparisons between two groups (death group vs. survival group) are performed using the Wilcoxon rank-sum test.

**Table 4 ijms-24-13857-t004:** Investigations into factors affecting outcomes by univariate and multivariate analyses.

	Univariate Analysis		Multivariate Analysis	
Background at Admission	Odds Ratio (95% CI)	*p*-Value	Odds Ratio (95% CI)	*p*-Value
Age	1.115 (1.022–1.215)	0.002	1.089 (0.995–1.229)	0.068
Sex	0.4 (0.109–1.461)	0.152		
BMI (kg/m^2^)	0.990 (0.862–1.127)	0.884		
SOFA score	1.46 (1.13–1.97)	0.0025	1.169 (0.819–1.704)	0.384
WBC count (×10^3^/μL)	0.993 (0.888–1.109)	0.895		
PLT count (×10^4^/μL)	0.996 (0.990–1.003)	0.214		
CRP (mg/dL)	1.042 (0.987–1.101)	0.137		
Cr (mg/dL)	1.413 (1.026–2.018)	0.035		
eGFR (mL/min/1.73 m^2^)	0.979 (0.948–1.001)	0.071		
T-bil (mg/dL)	1.155 (0.817–1.625)	0.375		
LDH (U/L)	1.000 (0.999–1.004)	0.084		
AST (U/L)	1.001 (0.999–1.004)	0.130		
ALT (U/L)	1.002 (0.998–1.006)	0.290		
UA (mg/dL)	1.071 (0.931–1.229)	0.322		
U.excretion of UA (mg/g·Cr)	0.138 (0.001–2.758)	0.267		
Lactate levels (mmoL/L)	1.367 (1.120–1.668)	0.0003	1.223 (0.993–1.602)	0.059
Blood XDH levels (ng/mL)	14.25 (2.51–80.79)	0.0002	8.839 (1.417–91.21)	0.018
Blood 8-OHdG levels (pg/mL)	0.9945 (0.987–1.001)	0.109		

Abbreviations: BMI, body mass index; SOFA, Sequential Organ Fairure Assessment score; WBC, white blood cells; PLT, platelet; CRP, C-reactive protein; Cr, Creatinine; eGFR, estimated glomerular filtration rate; T-bil, total bilirubin; LDH, lactate dehydrogenase; AST, aspartate aminotransferase; ALT, alanine aminotransferase; UA, uric acid; U.excretion of UA, urinary excretion uric acid; XDH, enzyme xanthine dehydrogenase; 8-OHdG, 8-hydroxy-2-deoxyguanosine.

## Data Availability

Data supporting the findings of this study are available from the corresponding author, J. Y., upon reasonable request.

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
