# Peer review of "Clinical Significance of Elevated Xanthine Dehydrogenase Levels and Hyperuricemia in Patients with Sepsis"

_ijms, 2023, doi:10.3390/ijms241813857_

Round 1
Reviewer 1 Report
This article is very creative and important. But the last sentence in the Abstract could be revised to make it perfect.
"Countermeasures against increased oxidative stress in sepsis may provide new therapeutic targets. " could be revised as :
"Countermeasures against increased oxidative stress in sepsis may provide new therapeutic strategies.
or
"Elevated oxidative stress in sepsis may be a new therapeutic target".
or
Management of increased oxidative stress in sepsis may be a new therapeutic direction ( or way ) .
Reviewer 2 Report
Dear Doctor Matsuoka!
I have an honor to review your manuscript ”Clinical Significance of Elevated Xanthine Dehydrogenase Levels and Hyperuricemia in Patients with Sepsis”.
Your work is devoted to the most important problem of studying the methods of prognosis in sepsis and has an unconditional scientific and applied, practical value.
Nevertheless, despite the great interest that your work has aroused, I must note several fundamentally important remarks that require your attention and correction.
First of all, I would like to note that the construction of a scientific article has been violated. Materials and methods should be used in the construction of a scientific article before the results and before the discussion.
See also materials on demand for the authors of this journal: “…all manuscripts must contain the required sections: Author Information, Abstract, Keywords, Introduction, Materials & Methods, Results, Conclusions, Figures and Tables with Captions, Funding Information, Author Contributions, Conflict of Interest and other Ethics Statements.” IJMS | Instructions for Authors (mdpi.com)
Line 36-37. Oxidative stress is defined as an imbalance between the production of reactive oxygen species (ROS) in the body and antioxidant mechanisms, leading to an overall tendency towards oxidation. But the generally accepted definition of oxidative stress sounds somewhat different, it was given by the same author whom you indicate in link 5, and it contains a fundamentally important point concerning violations of redox regulation, and not only the balance of oxidants and antioxidants and cellular damage: Oxidative stress is proposed to be considered as "an imbalance between oxidants and antioxidants in favor of oxidants, leading to a violation of redox signaling and control and/or molecular damage" (Sies, Jones, 2007).
I cannot agree with your statement that there are no markers for the diagnosis of oxidative stress, rather, there are no generally accepted criteria for oxidative stress, and methods of assessment differ - there are a lot of works dedicated to this problem.
Line 269. Description of the properties of the enzyme xanthine oxidoreductase should be given at the beginning,at introduction, not in the discussion.
Line 275 not O2- but O2-, please give in the text the chemical designations of oxygen and radicals to the existing rules.
At Line 367-379 you tell, “The XOR gene is activated by various factors, including interferon-gamma (IFN-É£), interleukins (IL) 1 and 6, glucocorticoid, hypoxia, and lipopolysaccharides”, but there are absolutely no references in the work to the fact that oxidative stress is accompanied by a violation of RedOx regulation, although the above example just refers to these violations. This point should be covered and discussed more deeply.
Line 391 I don't understand if you mean February 2018 or February 2019, if February 2019, then why the periods of the group recruitment were different, if February 2018, you just need to specify it.
Line 22 What specific statistical methods of analysis have you used here, it is necessary to indicate, despite the fact that there is a subsection "statistical analysis"
Line 486-488 your statement is controversial, since there is an opinion that uric acid is a two-faced Janus in the biochemistry of oxidative stress, and in high doses can have pro-oxidant properties. This point should be noted, there are many works on this topic, for example: in review (Hayden, Melvin R. and Tyagi, Suresh C. 2004. “Uric acid: A New Look at an Old Risk Marker for Cardiovascular Disease, Metabolic Syndrome, and Type 2 Diabetes Mellitus: The Urate Redox Shuttle”. Nutrition & Metabolism. (Lond.). 1: 10. https://doi.org/10.1186/1743-7075-1-10.) the authors, discussing the role of increased concentration of uric acid as a marker and risk factor for cardiovascular disease argue that the uric acid concentration in excess of normal values, as well as its concentration within the upper third of the physiological range should be considered as one of the multifactorial damaging incentives in relation to arterial walls and capillaries, leading to the development of endothelial dysfunction and vascular wall remodeling through OS.
There are some other works where the elevated level of uric acid is associated with high risk of cardiovascular events - Framingham study (Kannel, Castelli, McNamara 1967), and some other works where the contradiction between the expressed antioxidant properties of uric acid, and increase in risk of development of cardiovascular diseases at its high concentration were demonstrated (Alcaino et al. 2008, Sautin et al. 2007, Tziomalos et al. 2010). The idea you have told about the endothelial disfunction in sepsis in your work seems absolutely correct, but, please, make more accurate assessment about the role of uric acid in high concentrations as antioxidant.
Alcaino, Hernan; Greig, Douglas; Chiong, Mario; Verdejo, Hugo; Miranda, Rodrigo; Concepcion, Roberto; Vukasovic, José L.; Diaz-Araya, Guillermo; Mellado, Rosemarie; Garcia, Lorena; Salas, Daniela; Gonzalez, Leticia; Godoy, Ivan; Castro, Pablo and Lavandero, Sergio. 2008. “Serum Uric Acid Correlates with Extracellular Superoxide Dismutase Activity in Patients with Chronic Heart Failure”. European Journal of Heart Fail, Vol. 10, issue 7: 646-651. https://doi.org/10.1016/j.ejheart.2008.05.008.
Tziomalos, Konstantinos; Athyros, Vasilios G.; Karagiannis, Asterios and Mikhailidis, Dimitri P. 2010. “Pitfalls in the evaluation of uric acid as a risk factor for vascular disease”. The Open Clinical Chemistry Journal. Vol.3: 44-50. http://dx.doi.org/10.2174/1874241601003010044
Sautin, Yuri Y.; Nakagawa, Takahiko; Zharikov, Sergey and Johnson, Richard J. 2007. “Adverse Effects of the Classical Antioxidant Uric Acid in Adipocytes: NADPH Oxidase-Mediated Oxidative/nitrosative Stress”. American Journal of Physiology-Cell Physiology, Vol. 293, issue 2: C584-C596. https://doi.org/10.1152/ajpcell.00600.2006.
Reviewer 3 Report
Introduction
1. The text proposes the need for more sophisticated treatment of sepsis cases with new pathophysiological analyses using adequate markers. What are the current limitations in using existing markers or scores like SOFA in understanding sepsis pathophysiology?
2. The study aims to investigate the clinical significance of uric acid (UA) and related molecules for measurement. Have there been any previous studies that have explored UA's potential role as an antioxidant in sepsis, or is this a novel hypothesis?
3. Could you explain why UA was chosen as a potential marker of oxidative stress in sepsis? Are there specific attributes of UA that make it a suitable candidate for this role?
4. Reference (5) links oxidative stress to organ dysfunction. Can you elaborate on the mechanisms by which oxidative stress contributes to organ dysfunction in sepsis?
Results
1. Figure 5a-b presents a comparison of blood UA levels between the control group and the survival and death groups. Can you explain why there was a statistically significant difference in blood UA levels between the control group and the survival group on day 0, but not on subsequent days?
2. Could you explain the clinical relevance of the cutoff value for blood XDH levels at 1.38 ng/mL derived from the receiver operating curve (ROC) analysis in predicting outcomes (survival vs. death)?
Discussion
1. The text mentions that Xanthine oxidoreductase (XOR) exists in two forms: XDH and xanthine oxidase (XO). Could you explain the specific roles of XDH and XO in the context of sepsis and oxidative stress?
2. In this study, blood XDH levels were positively correlated with blood UA levels but negatively correlated with blood 8-OHdG levels. What are the possible implications of these correlations for understanding the relationship between XDH, UA, and oxidative stress in sepsis?
3. Are there any previous studies or evidence suggesting that XDH might play a role in sepsis outcomes, and how does your study contribute to the existing knowledge in this area?
4. The potential therapeutic effects of XOR inhibitors are discussed in chronic diseases. Considering the findings of your study, do you think XOR inhibitors could be explored as a potential treatment option for sepsis in the future?
5. What are the potential confounding factors or variables that could influence the relationship between blood XDH levels and sepsis outcomes, and how were these factors controlled for in the study?
6. The study focused on blood XDH levels, but it is mentioned that XOR can be released from various organs. Could you explain if there were any specific reasons for focusing on blood XDH levels and if measuring XOR levels in specific organs might provide additional insights?
7. Are there any implications of the study's findings for clinical practice in the management of sepsis patients, and are there any potential avenues for further research based on the results obtained?
8. The study suggests that blood XDH levels may be used as a pathophysiological biomarker to predict sepsis outcomes. Could you discuss the practical implications of this finding and the potential utility of XDH measurement in clinical settings?
9. The study discusses the positive correlation between blood XDH levels and SOFA scores, indicating sepsis severity. How does this relationship align with the current understanding of sepsis pathophysiology and what further research could be done to elucidate the link between XDH and disease severity?
10. The study highlights the positive correlation between blood XDH levels and cytokines like IL-6. Could you discuss how the results might contribute to our understanding of the inflammatory response in sepsis and its relationship with oxidative stress mediated by XOR?
11. The study focuses on XOR inhibitors and their potential therapeutic effects in chronic diseases. Given the findings related to XDH and sepsis outcomes, could you discuss if these inhibitors might also be explored for their potential in sepsis treatment?
12. In the text, the study discusses the potential impact of multiple factors on blood XDH levels in sepsis patients, including systemic inflammatory responses, oxidative stress, and specific pathogens. How might these factors interact with XDH levels, and what areas of research could further elucidate these complex relationships?
Methods
1. The text mentions the use of enzyme-linked immunosorbent assays (ELISA) to measure blood XDH and 8-OHdG levels. Could you provide more details about the reliability and accuracy of these assays and any potential sources of variability in the measurements?
2. The text mentions that the outcomes were evaluated as survival or death at the time of discharge. Were there any other outcome measures considered, such as length of ICU stay or overall hospital mortality?
3. The percentage change in blood UA and XDH levels from day 0 to days 1 and 3 was calculated. What were the reasons for choosing these specific time points, and were there any significant trends or patterns observed in the percentage changes for these biomarkers?
4. It is mentioned that urinary excretion of UA may be affected by hydration levels in the body. How were efforts made to control for hydration status in the analysis, and were there any significant correlations between hydration status and blood UA or XDH levels?
5. The study acknowledges some limitations, such as the relatively small sample size and the exclusion of patients unable to provide urine samples. How might these limitations affect the interpretation of the results, and do you believe they could impact the study's overall conclusions?
Minor editing of English language required
Round 2
Reviewer 2 Report
Dear Doctor Matsuoka!
I have an honor to review your revised manuscript ”Clinical Significance of Elevated Xanthine Dehydrogenase 2 Levels and Hyperuricemia in Patients with Sepsis” – Version 2.
Your work is devoted to problem of studying the methods of prognosis in sepsis and has an unconditional scientific and applied, practical value.
I see that after deep revision the construction of your scientific article has been changed and the article is brought to the generally accepted rules for compiling a scientific text. All notes have been carefully studied and the necessary adjustments have been made.
I have only one new note – please correct the Reference 61 you have pointed at line 510 (Yuri et. al (61) to Sautin et. al, as correctly pointed at line 902 (Yuri is the first name and Sautin the Family name of investigator).
Reviewer 3 Report
Paper can be accepted in its present form.
